# Exploring the Cause of Diarrhoea and Poor Growth in 8–11-Week-Old Pigs from an Australian Pig Herd Using Metagenomic Sequencing

**DOI:** 10.3390/v13081608

**Published:** 2021-08-13

**Authors:** Tarka Raj Bhatta, Anthony Chamings, Soren Alexandersen

**Affiliations:** 1Geelong Centre for Emerging Infectious Diseases, Geelong, VIC 3220, Australia; trbhatta@deakin.edu.au (T.R.B.); anthony.chamings@deakin.edu.au (A.C.); 2School of Medicine, Deakin University, Geelong, VIC 3220, Australia; 3Barwon Health, University Hospital Geelong, Geelong, VIC 3220, Australia

**Keywords:** metagenomic sequencing, disease, viruses, bacteria, abundance, phylogenetic analysis

## Abstract

Diarrhoea and poor growth among growing pigs is responsible for significant economic losses in pig herds globally and can have a wide range of possible aetiologies. Next generation sequencing (NGS) technologies are useful for the detection and characterisation of diverse groups of viruses and bacteria and can thereby provide a better understanding of complex interactions among microorganisms potentially causing clinical disease. Here, we used a metagenomics approach to identify and characterise the possible pathogens in colon and lung samples from pigs with diarrhoea and poor growth in an Australian pig herd. We identified and characterized a wide diversity of porcine viruses including RNA viruses, in particular several picornaviruses—porcine sapelovirus (PSV), enterovirus G (EV-G), and porcine teschovirus (PTV), and a porcine astrovirus (PAstV). Single stranded DNA viruses were also detected and included parvoviruses like porcine bocavirus (PBoV) and porcine parvovirus 2 (PPV2), porcine parvovirus 7 (PPV7), porcine bufa virus (PBuV), and porcine adeno-associated virus (AAV). We also detected single stranded circular DNA viruses such as porcine circovirus type 2 (PCV2) at very low abundance and torque teno sus viruses (TTSuVk2a and TTSuVk2b). Some of the viruses detected here may have had an evolutionary past including recombination events, which may be of importance and potential involvement in clinical disease in the pigs. In addition, our metagenomics data found evidence of the presence of the bacteria *Lawsonia intracellularis*, *Brachyspira* spp., and *Campylobacter* spp. that may, together with these viruses, have contributed to the development of clinical disease and poor growth.

## 1. Introduction

Porcine enteric diseases resulting in diarrhoea or intestinal pathology and poor growth performance cause huge economic losses around the world [1,2,3,4,5,6,7,8]. It has been suggested that various etiological agents acting either individually or synergistically including viruses, bacteria, and parasites are responsible for clinical consequences such as diarrhoea and poor growth in pigs [9,10]. Viruses, including picornaviruses such as porcine sapelovirus (PSV), porcine enteroviruses (PEV), and porcine teschoviruses (PTV), have been identified in association with various disorders, including diarrhoea, polioencephalomyelitis, respiratory distress, dullness, skin lesions, pyrexia, and flaccid paralysis [11,12,13,14,15]. Porcine astrovirus (PAstV) has also been associated with gastrointestinal, respiratory, and neurological disorders [16,17,18]. Porcine circovirus type 2 (PCV2) has been associated with a number of diseases/clinical syndromes together called porcine circovirus associated diseases (PCVAD) [19], including postweaning multisystemic wasting syndrome (PMWS) and porcine dermatitis and nephropathy syndrome (PDNS) [20,21]. Some of the clinical sign of PMWS include growth retardation, enlarged lymph nodes, and wasting [22]. It has been shown that co-infection of PCV2 with porcine parvoviruses (PPVs), porcine bocaviruses (PBoVs), and torque teno sus viruses (TTSuVs) can be associated with porcine respiratory diseases complex (PRDC) [23] and PMWS [24]. Bacteria such as *Lawsonia intracellularis* [25,26,27], *Brachyspira* spp. [28,29,30], *Campylobacter* spp. [31], *Escherichia coli*, and *Salmonella* spp. [32] have also been found to be associated with diarrhoea and poor growth in pigs. Culture and targeted polymerase chain reaction (PCR) can be performed for the detection of these bacteria [33,34]. Detection of specific porcine viruses known or likely to be present, e.g., swine influenza A virus (swIAV) [35], PSV [36], PEV [37], PTV [38], PAstV [39], PBoV [40,41], PCV2 [42], and many others, can also be performed using targeted PCR assays. However, targeted detection may not be suitable for investigating co-infection with multiple pathogens, or if there is an involvement of novel or variant viruses and other pathogens [43].

Next generation sequencing (NGS) technology is a very useful tool for the metagenomic detection and characterisation of multiple, unexpected, novel, or unidentified viruses from healthy and diseased individuals [44,45,46,47]. Various independent studies have used NGS for the detection of diverse groups of viruses in pig samples including PEV, PAstV, PCV2, atypical porcine pestivirus (APPV), PBoV, TTSuVs, and many others [47,48,49,50]. Similarly, different pathogens were also identified in bat [51], dog [44,52], bird [45,46,53], rodent [54], and human samples [55,56] using NGS technology.

Previously we have used our metagenomics approach for the detection and characterisation of various viruses (both RNA and DNA) in dogs suffering from diarrhoea [44] and in faecal samples from wild birds [45,46] and humans [45]. The study described here focused on the metagenomic detection and characterisation of pathogens from a single pig herd located in South Eastern Australia that was experiencing diarrhoea and poor growth in growing pigs from eight weeks of age onwards.

## 2. Materials and Methods

### 2.1. Clinical History and Sample Collection

The affected pig herd was located in South Eastern Australia. The farm produced and recruited gilts internally but did use commercially bought local boar semen as part of their breeding program. There were no recent introductions of pigs to this farm; the last introduction being 10 years prior from a genetic herd in Queensland, Australia. Progeny pigs were not vaccinated against PCV2 or any other pathogens. When visited by the veterinarian in late August 2018, approximately eight to twelve pigs from a total of 52 pigs in a pen had started showing relatively severe signs of gastrointestinal disease/diarrhoea from approximately eight weeks of age. A few pigs had died while some were skinny and not growing well compared to other pigs in the pen. Treatment was initiated, including Tylosin injection (Tylan, Elanco) which resulted in no noticeable improvement. The concentration of Tylosin in the feed was increased from 50 to 200ppm with a slight beneficial effect. The diet was also supplemented with 100ppm Olaquindox as a growth promoter. The pigs were then injected with 10mg/kg Engemycin (Oxytetracycline), a broad spectrum antibiotic active against both Gram-positive and Gram-negative bacteria, which resulted in some clinical improvement in some of the pigs, but the problem persisted to eleven weeks of age. At eleven weeks, (mid-September 2018), two (Pig 45 and Pig 46) of the affected pigs in the pen were euthanised for post-mortem examination by the attending farm veterinarian. Gross findings included mild proliferative lesions in the small and large intestines. The colon was enlarged and there was some inflammation of the colon mucosa along with excessive mucus. Proliferative enteropathy (*Lawsonia intracellularis*) was suspected, and samples were submitted for histopathology along with culture for *Salmonella* and *Brachyspira* spp. at a veterinary diagnostic laboratory. In addition, fresh samples of surplus colon and lung tissues from the two pigs were forwarded overnight to the Geelong Centre for Emerging Infectious Diseases (GCEID) in an Esky with ice packs to perform a metagenomics investigation using NGS. Intestinal and respiratory tissues were chosen as these systems are common sites of virus excretion. Upon receipt, swabs were taken from each of the mucosal epithelial surfaces of the colon and multiple sites within the lung tissues from each pig, and the swabs were placed in universal transport medium (UTM). Aliquots of the tissue suspensions were frozen at minus 80 °C until processed further.

### 2.2. Enrichment, Nucleic Acid Extraction, cDNA Synthesis and Non-Targeted Amplification

For NGS, each colon and lung sample from each pig were processed separately, and small particle-protected nucleic acids enriched using a previously described protocol optimised in our laboratory with some modifications [44,45,57]. Briefly, the samples were homogenised at 25 Hz for 2 min using a TissueLyzer (Qiagen, Hilden, Germany) followed by centrifugation at 17,000× *g* for 3 min and then filtered using a 0.8 µm polyether sulfone (PES) spin-column filter. The samples were then ultracentrifuged at 178,000 g for 1 h. Pelleted small particles were then treated with benzonase and micrococcal nuclease for 2 h to enrich for nucleic acids being protected in the particles [44]. The ultracentrifuged and nuclease treated samples were then treated with 200 µM propidium monoazide (PMAxx) for 30 min in a PMA-Lit LED photolysis device to cross-link nucleic acids not protected in particles, as indicated by the manufacturer (Biotium, Fremont, CA, USA) [57]. Finally, total nucleic acids were extracted using the QIAamp Viral RNA Mini Kit (Qiagen, Hilden, Germany). For cDNA synthesis and DNA amplification, an initial RNA denaturation step of 95 °C for 3 min followed by snap-cooling in an ethanol–ice bath (minus 20 °C) was carried out before using the SeqPlex RNA Amplification Kit (Sigma, St. Louis, MO, USA), as per the manufacturer’s protocol [44,45]. Bioanalyzer and high-sensitivity DNA chips (Agilent, Waldbronn, Germany) were used to check the quantity and quality of the amplified product.

### 2.3. Library Preparation, Next Generation Sequencing (NGS), and Detection of Different Virus Sequences

Library preparation was performed using the Ion Plus Fragment Library Kit and IonXpress Barcode Adapters 1-96 Kit (Thermo Fisher Scientific, Waltham, MA, USA). The libraries were quantified using the Ion Library TaqMan Quantitation Kit (Thermo Fisher Scientific, Waltham, MA, USA) and then pooled. Sequencing was performed using Ion 530 chips and an Ion Torrent S5XL System (Thermo Fisher Scientific, Waltham, MA, USA). Each of the pooled individual samples generated approximately 2.8 to 6.9 million reads. Generated sequence reads were compared to a local database of virus sequences downloaded from GenBank (December 2018) using BLASTN and BLASTX [58,59] queries with an e-value cut-off score of 1 × 10^−10^. Contigs were generated using the AssemblerSPAdes 5.12.0.0 plugin on the Ion Torrent Server and were compared to the virus reference sequences in NCBI GenBank using BLASTN and BLASTX to identify virus sequences. NGS reads were mapped to the contigs matching virus sequences using the TMAP plugin [60] to determine the coverage depth. The TMAP plugin [60] from the Ion Torrent suite was also used for mapping the NGS reads using selected virus sequences from NCBI GenBank that were identified as most similar to the viruses in the pig samples. Consensus sequences were generated based on the number of high-quality reads (initially MAPQ >20) matched to the reference viruses and coverage/abundance of the reads across the reference genomes.

Near full or partial consensus sequences were generated from the mapped sequences using Integrative Genomics Viewer software (IGV) (Broad Institute, MA, USA) [61] with a mapping quality score of 64 or higher and a coverage depth of at least two with further specific details noted in the results below. Generated sequences were compared with the online NCBI GenBank database using BLASTN [58,59], and the closest, and additional related or representative sequences from around the globe, were selected for further analyses. Selected sequences were aligned using Clustal-W [62] and phylogenetic trees, created using the maximum likelihood (ML) method with the best fitting model, as determined by MEGA 7 [63]. The robustness of different phylogenetic nodes was assessed using 1000 bootstrap replicates for nucleotide sequences.

### 2.4. Similarity Plot (SimPlot) Analysis of Identified Virus Sequences

Similarity plot (SimPlot) was used to look for possible recombination between our virus sequences obtained from the pig samples and related sequences from NCBI GenBank. A 200 nucleotide sliding window at 20 nucleotide intervals and the F84 distance model with maximum likelihood method was used. Percentage identities at each analysis point were plotted on a line chart [64].

### 2.5. Mapping of Bacterial and Antimicrobial Resistance (AMR) Gene Reads

Although our NGS method enriched for virus particles, the final NGS reads also includes some reads derived from other nucleic acids, e.g., from ribosomes, mitochondria, and bacterial or host genomes [45]. Consequently, mapping of NGS reads to bacterial DNA was performed using CLC Workbench version 20.0 Microbial Genomics Module (Qiagen, Hilden, Germany) with the included “Taxonomic Profiling” tool and the microbial genome database from January 2020. Mapping of reads to the 16S ribosomal RNA microbial databases was also done on the Thermofisher Ion Reporter website, using the setting of at least two identical reads mapping over at least 60 nucleotides (60-60-2). Additionally, a BLASTN query with an e-value cut-off score of 1 × 10^−10^ was performed against the refseq ribosomal (5S, 16S and 23S) database from the NCBI GenBank genetic sequence database (downloaded September 2019). Finally, CLC and the Microbial Genomics Module were used to map against the CLC antimicrobial resistance (AMR) gene databases using CLC assembled contigs and the tool “Find resistance with nucleotide database”.

## 3. Results

NGS was conducted on swab samples taken from fresh chilled samples of surplus colon and lung tissues from two pigs experiencing diarrhoea and poor growth at 8–11 weeks of age. Based on the initial BLASTN and BLASTX analyses of the NGS reads and their abundance, both RNA and DNA viruses representing 0.07–1.37% of the total sequenced reads were identified in samples from both pigs (Table 1). RNA viruses identified were picornaviruses: including porcine sapelovirus (PSV), enterovirus G (EV-G), porcine teschovirus (PTV), and a porcine astrovirus (PAstV). Single Stranded DNA viruses identified were parvoviruses: porcine bocavirus (PBoV), porcine parvovirus 2 (PPV2), porcine parvovirus 7 (PPV7), porcine bufa virus (PBuV), porcine adeno-associated virus (AAV), porcine circovirus type 2 (PCV2), and torque teno sus viruses (TTSuVk2a and TTSuVk2b).

Despite employing a virus enrichment method to remove/degrade much of the non-virus nucleic acids, between 0.30–1.29% of the total NGS sequenced reads were mapped to bacterial chromosomal or plasmid DNA and a small number of reads mapped to the corresponding bacterial ribosomal RNA in each of the samples from both pigs (Appendix A). These reads mapped to a number of different bacterial species, but of particular interest were NGS reads mapping to pathogenic bacteria which are known to cause diarrhoea and poor growth in pigs (e.g., *Lawsonia intracellularis*, *Brachyspira* spp., and *Campylobacter* spp.).

Conventional diagnostic methods (histopathology and bacterial culture) detected *Lawsonia intracellularis* and *Brachyspira* spp. The NGS metagenomics sequencing confirmed the presence of these agents and was able to detect sequences from at least 12 viruses (Table 1) that were not tested for by conventional diagnostics methods. Nine of these viruses were of particular interest because of either their relatively high abundance and/or their recognised potential to cause clinical signs consistent with those observed in the herd. These viruses included porcine sapelovirus (PSV), enterovirus G (EV-G), porcine teschovirus (PTV), porcine astrovirus (PAstV), porcine bocavirus (PBoV), porcine parvovirus 2 (PPV2), porcine circovirus type 2 (PCV2), and torque teno sus viruses (TTSuVk2a and TTSuVk2b) (see below).

### 3.1. Porcine Sapelovirus (PSV)

Sequences belonging to PSV made up between 0.02 and 1% (Table 1) of total sequenced reads from the tissue samples. Four and six partial consensus sequences representing different regions of at least two genetically distinct PSVs were assembled from the mapped reads using a mapping quality of 80–90 and coverage of 2–5638 from the swabs of Pig 45 and Pig 46, respectively (Appendix A). These assembled sequences each included the full sequence of the virus capsid structural proteins (VP4, VP2, VP3, and VP1), the full sequence of the non-structural virus proteins P2 (2A-2B-2C), and a partial sequence of the non-structural proteins P3 (3A-3B-3C (but not 3D)) of PSV except for sapelovirus sequenced in the Pig 46 lung sample.

Sequence comparison of the VP1 region has been used for the genotyping of several picornaviruses as this is one of the most variable capsid proteins and important in inducing specific host immunity [65]. In the case of sapelovirus, the VP1 sequence is 844 nt long and encodes for a 281 amino acid (aa) long protein. Two VP1 sequences from the colon and lung of Pig 45 were 100% identical with each other and were also 100% identical to the partial VP1 sequence from the lung sample of Pig 46. The VP1 sequences from the colon and lung samples of Pig 45 were only 80.2% identical with the closest PSV sequence (LC508229-PSapV/22-B/Zambia-2018) from Zambia and were only 74.6% identical with the VP1 sequence obtained from the colon sample from Pig 46, indicating that different porcine sapeloviruses were circulating on the farm and were possibly co-infecting Pig 46. The VP1 sequence from Pig 46 colon was 88.9% identical with the closest PSV sequence (MK378907-PSapV/GX04 C1/China-2017) from China. Interestingly, the VP1 sequence from the Pig 46 colon sample has a 21 (ACAGGTTTCTACCCTGCAACC) nucleotide (7 amino acids: TGFYPAT) insertion/repeat towards the end of VP1, while this is absent in the sapelovirus sequences from the colon and lung of Pig 45 and that from the lung sample of Pig 46. The insertion has been reported for some other sapelovirus and other picornavirus sequences as “protein of unknown function (DUF3724)” and is a short domain of approximately 20 aa in length in which a completely conserved tyrosine (Y) amino acid residue which may be functionally important [66].

Pairwise comparison of the 1827 nt long (609 aa) VP4, VP2, and VP3 (VP4-VP3) sequence of the capsid region and the 3216 nt long (1072 aa) P2-P3 (but not 3D) sequence of the non-structural region with the reference sapelovirus sequences from NCBI GenBank was conducted. VP4-VP3 sequences of Pig 45 colon and lung and Pig 46 lung swab were 100% identical with each other. These three sequences were 85.2% identical at the nt level with the closest PSV sequence LC425396-PSV/DeTk2-2/Japan-2015 from Japan and were only 81.7% identical with the VP4-VP3 sequence from the Pig 46 colon swab. KJ821019-PSV/KS04105/KOR-2004 from Korea was the closest sequence with 88.1% identity at nt level with VP4-VP3 sequence from the Pig 46 colon swab. The non-structural P2-P3 region of the PSVs from Pig 45 colon and lung were 100% identical with each other and were only one nucleotide different from the partial P2-P3 sequences of Pig 46 lung. The P2-P3 sequences from Pig 45 colon and lung were 96.6% identical with P2-P3 sequences from Pig 46 colon. All the P2-P2 sequences from this study were 89% to 89.1% identical at nt level with the closest reference, MK378928-PSV/HLJ01 C1/China-2017 from China. The PSV sequences from Pig 45 colon and lung and Pig 46 lung samples were therefore reasonably similar in their non-structural regions to Pig 46 colon sample but had a more different capsid region between nucleotides 400–3000 (Figure 1). The different capsid regions of these two viruses may have evolved through recombination or possibly immune selection of different capsid genotypes/serotypes whereas the non-structural region was less divergent.

### 3.2. Porcine Enterovirus G (PEV-G)

Sequences corresponding to PEV-G made up 0.02 to 0.35% of the sequenced reads from the colon and lung tissue samples from the two pigs (Table 1). Two identical complete coding sequences of PEV-G (Pig 45 colon; 6993 nt (coverage 2–3448) and Pig 45 lung; 7189 nt (coverage 2-1686)) representing a single porcine enterovirus-G were obtained from Pig 45 at mapping quality of 80. However, 15 partial PEV-G sequences (Appendix A) representing multiple porcine enteroviruses-G were obtained from the colon sample of Pig 46 and five partial consensus sequences of PEV-G were obtained from the lung sample of Pig 46 (Appendix A).

As the VP1 sequence determines the genotype of EV-G, and the criteria to distinguish/separate one genotype from another is a variation of VP1 nucleotide sequence of more than 25% [49,67], a pairwise comparison of the VP1 nucleotide sequences was conducted. We obtained four full sequences of the 729 nt long VP1 capsid of PEV-G, one from each of the lung and colon tissue samples of Pig 45 and Pig 46 and these were used for phylogenetic analysis using thirty nine reference sequences belonging to EV-G1 to EV-G20 [49,68,69] from https://www.picornaviridae.com/sg3_ensavirinae/enterovirus/ev-g/ev-g.htm (accessed on 18 March 2021) and from NCBI GenBank. The two VP1 sequences from the colon and lung from Pig 45 (MZ515508 and MZ515509, respectively), and one VP1 sequence from the lung sample of Pig 46 (MZ515525) were 100% identical with each other. These three sequences were only two nucleotides different from one of the VP1 sequences (MZ515510-EV-G-VP1-AUS-2018) from the colon sample of Pig 46. These four VP1 sequences clustered with EV-G1 genotype (Figure 2) and were closest (~80.1–80.4% identical) to KT265971-EV-G1/734141/ThanhBinh-Vietnam-2012 from Vietnam by nucleotide pairwise comparison. The other partial (363 nt long) VP1 sequence (MZ515511-EV-G-VP1-AUS-2018) from the colon of Pig 46 was most similar to the EV-G4 reference genotype and was only 69% identical with the PEV-G VP1 sequence mentioned above (MZ515510-EV-G-VP1-AUS-2018). The EV-G4 like VP1 sequence from Pig 46 colon sample was closest (~81% identical at nt level) to MF113307-EV-G4/F22-1/03-05-GER-2013 from Germany. Six near full length sequences of VP4, VP2, and VP3 (VP4-VP3) capsid region of PEV-G were obtained from Pig 45 colon and lung and Pig 46 colon samples (Appendix A) and used for phylogenetic analysis. Sequences most similar to genotypes EV-G1 and EV-G4 were again identified, as was one sequence most similar to genotype, EV-G2 and another to genotype EV-G9 (Appendix A) both from the Pig 46 colon sample.

The non-structural P2-P3 sequences from the tissues of Pig 45 and lung of Pig 46 were most similar to the reference sequences from the EV-G1 genotype whereas sequences closest to EV-G1 and EV-G2 genotypes were obtained from the Pig 46 colon samples. No P2-P3 sequences belonging to the genotypes EV-G4 and EV-G9 were identified. Despite only obtaining partial sequences of PEV-G from Pig 46, there was clearly a number of genetically different EV’s present in the tissues of this pig at the time of sampling, although some were at very low abundance and therefore unlikely to be actively replicating at the time of sampling.

### 3.3. Porcine Teschovirus (PTV)

Sequences corresponding to PTV with 0.007 to 0.024% (Table 1) abundance of total sequenced reads were obtained. Four and seven partial consensus sequences representing different regions of porcine teschovirus (PTV) were assembled from the mapped reads using a mapping quality of 80 and coverage of 2–20 from the colon swabs of Pig 45 and Pig 46, respectively (Appendix A). No sequences of PTV were obtained from either of the pig lung samples.

To genotype the PTV sequences in the sample, we aligned the sequences we identified belonging to the VP1 gene sequence with the reference PTV sequences as it is considered a highly variable region containing neutralizing epitopes for differentiation of serotypes of the PTV [71,72,73]. A 610 nt long partial VP1 sequence (MZ515535-PTV-VP1-AUS-2018) encoding for 203 aa from the colon sample of Pig 46 was used for pairwise comparison and was found to belong to the PTV3 genotype with 82.2% nucleotide identity with the nearest PTV sequence, GQ293230-PTV3/Vir 1925/02/GER-2002 from Germany. Another 292 nt long partial VP1 sequence (MZ515534-PTV-VP1-AUS-2018) encoding 97 aa from the colon sample of Pig 46 (Appendix A) was found to belong to the PTV2 genotype with around 84.2% nucleotide identity with its nearest sequence AY392533-PTV2/DS 183/93/GER-2003 from Germany. Our two VP1 sequences representing PTV3 and PTV2 genotypes were only around 65% identical with each other at the nucleotide level and indicated the presence of at least two different genotypes/serotypes of PTV present in the colon of Pig 46.

### 3.4. Porcine Astrovirus (PAstV)

Sequences corresponding to PastV were only identified in the colon sample of Pig 46 and comprised 0.02% (Table 1) of the total sequenced reads in this sample. As a result, we were only able to generate 10 partial sequences of porcine astrovirus (PastV) from the NGS reads of the Pig 46 colon sample. The number of reads were much fewer than for PSV, PEV, PTV, and PBoV (Table 1), and therefore we were only able to generate low coverage consensus sequences of PAstV reads using a mapping quality threshold of 64 and above (Appendix A).

The ORF2 capsid region of AstV is the most genetically variable region of the genome [74] as it contains neutralizing epitopes that form the capsid protein including the spike domain [75,76]. The 1122nt (374aa) long partial ORF2 sequence (MZ515541-PAstV-ORF2-AUS-2018) was used for pairwise comparison and found to be most similar to the PAstV4 genotype and also most similar (81.2% identical at nt level) with the reference sequence LC201607-PAstV4-JPN-HgTa2-1-2-Japan-2015 from Japan. Another partial ORF2 (MZ515544-PAstV-ORF2-AUS-2018) sequence of 484 nt long was also found to be most similar to PAstV4 genotype and was closest (75.4% identical at nt level) to the reference sequence KU764486-PAstV4-15-12-USA-2015 from the USA. These two capsid sequences shared a 271 nt long overlapping region, however they were only 59.1% identical at nt level indicating that these two sequences belonged to two distinct PoAstV4 in the colon of Pig 46.

One 869 nt long partial ORF1a sequence (MZ515542-PAstV-ORF1a-AUS-2018) was obtained from the Pig 46 colon swab and was most similar (89.3% (776/869) nucleotide identity) to MK460231-PAstV4/CHN/WG-R2-China-2017 from China which belonged to PAstV genotype 4. Another 281 nt long ORF1a sequence (MZ515548-PAstV-ORF1a-AUS-2018) was obtained from the same Pig 46 colon sample which was only 75.1% (211/281) identical to our first ORF1a sequence MZ515542-PAstV-ORF1a-AUS-2018. It was therefore evident that there were at least two different PAstV4 viruses within the colon sample from Pig 46, with two different PAstV4 capsid sequences and two different PAstV4 ORF1a sequences present.

### 3.5. Porcine Bocavirus (PBoV)

Sequences corresponding to PBoV were identified in both the colon and lung samples from both Pig 45 and 46 and made up 0.002 to 0.225% (Table 1) of the total sequenced reads. Twenty-two partial consensus sequences representing different regions of multiple genetically distinct porcine bocavirus 3 (PBoV3) were assembled from the mapped reads using a mapping quality of 64 and above and coverage of 2–888 from the swabs of Pig 45 and Pig 46 (Appendix A).

Phylogenetic classification of parvoviruses is performed using a relatively conserved region of NS1 [77,78,79,80]. Two 1278 nt long partial NS1 sequences from the Pig 45 lung (MZ544035-PBoV-NS1-AUS-2018) and Pig 46 colon (MZ544039-PBoV-NS1-AUS-2018) samples were used for pairwise sequence comparison and phylogenetic analysis along with 21 other reference sequences representing different clusters [77,78,79,80] of PBoV3 that were chosen from NCBI GenBank database. The NS1 sequence obtained from the Pig 45 lung sample was most similar to virus sequences belonging to the PBoV3 cluster D [77,78,81,82] (Figure 3) with KF025384-PBoV/MN154-1/USA-2011 as the closest (~97.3% identical at nt level) reference sequence from the USA. However, the NS1 sequence from the colon of Pig 46 was closest (~92.4% identical at nt level) to KF360033-PBoV/swBoV CH437/China-2012 from China (Figure 3), which was proposed as belonging to a novel PBoV3 cluster [83] (here tentatively designated PBoV3 cluster F/?). These two NS1 sequences from the Pig 45 lung and Pig 46 colon swabs were only ~80.1% identical at nucleotide level with each other.

One partial 1193 nt long VP1 sequence from Pig 45 lung sample and two VP1 partial sequences each 1193 nt long from Pig 46 colon sample were created and used for pairwise sequence comparison. It was found that VP1 sequence from Pig 45 lung sample (MZ544034-PBoV-VP1-AUS-2018) was closest (90.6% nucleotide identity) to MN747338-PBoV/GXHX2017-3/China-2017 from China. However, the two VP1 sequences (MZ544041-PBoV-VP1-AUS-2018 and MZ544040-PBoV-VP1-AUS-2018) from Pig 46 colon sample were 79.4% identical at the nucleotide level from each other and were closest (78.5–97% nucleotide identity) to MT321514-PBoV/GDZ1/China-2018 from China. Overall, there are at least two genetically distinct PBoVs circulating within the colon of individual Pig 46 and multiple genetically distinct PBoVs circulating on the farm.

### 3.6. Porcine Parvovirus 2 (PPV2)

In addition to the porcine bocaviruses described above, other single stranded DNA parvoviruses identified were PPV2, PPV7, PBuV, and AAV. Sequences corresponding to PPV2 comprised 0.0001 to 0.0279% of the total sequenced reads from both Pig 45 and 46 lung samples. Two partial consensus sequences representing different regions of PPV2 were obtained from Pig 45 lung sample whereas one partial sequence representing VP1 region of PPV2 was obtained from Pig 46 lung sample (Appendix A).

A 1494 nt long NS1 sequence (MZ544051-PPV2-NS1-AUS-2018) from the Pig 45 lung sample was used for pairwise comparison and phylogenetic analysis with 17 other reference sequences from NCBI GenBank. Our NS1 sequence was found to be closest (~95.4% nucleotide identity) to reference MG345014-PPV2/S4/China-2017 from China but made its own separate branch as shown in Appendix A. From the Pig 45 lung sample, a 2982 nt long partial VP1 sequence (MZ544050-PPV2-VP1-AUS-2018) was obtained and codes for a 993 amino acid long sequence. On pairwise comparison, our VP1 sequence was closest (~93.7% identical at nt level) to reference KC687097-PPV2/64-PL/Poland-2011 from Poland. Moreover, our partial VP1 sequence (PL46-BC-25-PPV2-3622-3913-AUS-2018) from Pig 46 lung sample was 100% identical with our VP1 sequence from the Pig 45 lung sample. Like our NS1 sequence, our VP1 sequence also formed its own separate branch in the phylogenetic analysis (Appendix A), indicating a more distant relationship with other known PPV2 references and also the possibility of a distinct genotype.

### 3.7. Porcine Circovirus Type 2 (PCV2)

Although in very low abundance (only 0.00003 to 0.0007% of total sequenced reads), we were able to assemble four partial consensus sequences of porcine circovirus type 2 (PCV2) from the Pig 46 colon sample. The number of reads was less than for the other viruses described above, but we were interested in this virus because of its recognized ability to interact with other pathogens and its possible role in immunosuppression, therefore potentially making pigs susceptible to other infections [85]. The consensus sequences obtained included 627 nt (coverage 1–6) and 161 nt (coverage 1–2) long segments of the ORF1 region and 451 nt (coverage 1–3) and 100 nt (coverage 1–4) long segments of the ORF2 region. These were assembled from the mapped reads using a mapping quality threshold of 64 and above and have been deposited in the NCBI Sequence Read Archive (SRA) under SRA accession PRJNA745489.

The ORF2 region of the PCV2 genome is the most variable region and is also the most suitable region for performing phylogenetic studies [86]. The 451 nt long partial ORF2 sequence (PC46-BC-26-PCV2-ORF2-AUS-2018) from the Pig 46 colon sample was used for pairwise sequence comparison and phylogenetic analysis along with 18 other reference sequences representing different genotypes of PCV2 chosen from the NCBI GenBank. The 451 nucleotide long ORF2 sequence from this study fell into the PCV2b genotype cluster with only 1 nucleotide different from the nearest sequences from USA, Canada, and Denmark as shown in Appendix A. The 100 nucleotide long ORF2 partial sequence (coverage 1-4) also paired with 100% identity with the nearest sequences from USA, Canada, and Denmark (an overall identity of 99.82% if the 451 nt and 100nt ORF2 sequences are considered together) and also fell within the PCV2b genotype cluster indicating our two partial ORF2 sequences may have come from the same virus.

The 627 nt long partial ORF1 sequence (PC46-BC-26-PCV2-ORF1-AUS-2018) was used for pairwise sequence comparison and phylogenetic analysis with 18 other reference sequences as mentioned above. This analysis suggested that the partial ORF1 section of our PCV2 sequence from Pig 46 colon swab fell into the PCV2a genotype with the nearest reference sequences being MK504415-PCV2/KSU-KS-2017-PCV2-38/USA-2017 and KX828217-PCV2/KU-1614/KOR-2016 with 100% nucleotide identity as shown in Appendix A. In addition, the 161 nt long (coverage 1–2) ORF1 partial sequence also shared 100% identity with the two same reference sequences from USA and Korea, indicating that our two partial ORF1 sequences may have come from the same virus. The finding that the ORF1 region from the Pig 46 colon is of PCV2a genotype while the ORF2 region is of PCV2b genotype may indicate that two different PCV2 genotypes may be circulating together in the colon sample of Pig 46 or alternatively that it is a single PCV2a/b recombinant PCV2 [87].

### 3.8. Porcine Torque Teno Sus Virus (TTSuV)

Sequences representing two species of genus *Kappatorquevirus*, namely TTSuVk2a (0.0004–0.0119% abundance of total sequenced NGS reads) and TTSuVk2b (0.0058-0.0249% abundance of total sequenced NGS reads) were obtained from both pigs (Table 1). Nine partial consensus sequences (117–1276 nt long) of TTSuVk2a representing different ORFs (Appendix A) were assembled from the mapped reads using the mapping quality threshold of 90 from Pig 45 colon and lung and Pig 46 lung samples. One complete (2565 nt long) coding sequence (MZ544061-TTSuVk2b-AUS-2018) of TTSuVk2b was obtained from the Pig 45 lung sample. Four additional partial consensus sequences of TTSuVk2b of 203–1177 nt long representing different ORFs from Pig 45 colon and Pig 46 lung swabs were also obtained (Appendix A).

After pairwise comparison of our two partial TTSuVk2a sequences, MZ544058-TTSuVk2a-AUS-2018 and MZ544057-TTSuVk2a-AUS-2018, from the Pig 45 lung sample with the reference sequences in the NCBI GenBank, MN272073-TTSuVk2a/TTV_LV06/Brazil-2016 from Brazil was found to be one of the closest sequences with 97% (840/866) and 95.22% (1215/1276) identity at nt level, respectively. All the other partial sequences of TTSuVk2a obtained from Pig 45 colon and Pig 46 lung samples were identical with the TTSuVk2a sequences from the Pig 45 lung sample.

Pairwise comparison of the 2565 nt long TTSuVk2b sequence (MZ544061-TTSuVk2b-AUS-2018) from the Pig 45 lung sample with reference sequences in the NCBI GenBank, identified MT671970-TTSuVk2b/pork147/Brazil-2018 from Brazil as the closest sequence with 99.92% (2551/2553) identity at nucleotide level with only two nucleotides different (A to C at 64, A to G at 923). The nucleotide positions mentioned here were given on the basis of our TTSuVk2b sequence MZ544061-TTSuVk2b-AUS-2018. In addition, our TTSuVk2b sequence had a 12 nucleotides (TAAAAAGACAAC) insertion at our nucleotide position 2517–2528, which codes for an additional four amino acids (QIKR) towards the 3′end of our TTSuVk2b sequence as compared to the sequence from Brazil (MT671970-TTSuVk2b/pork147/Brazil-2018). Other sequences also shared high nucleotide identity with our sequences, with a 1384 nt long partial TTSuVk2b sequence (MN272088) from Brazil in 2016 being 99.78% identical with our TTSuVk2b sequence, while another 707 nt long partial TTSuVk2b sequence (MK378015) from China in 2017 was 99.86% identical with our TTSuVk2b sequence from the lung of Pig 45. All the partial sequences of TTSuVk2b obtained from Pig 45 colon and Pig 46 lung samples were identical with the TTSuVk2b sequence from the lung of Pig 45. We also observed the same 12 nt insertion in TTSuVk2b sequence from the Pig 45 colon sample whereas we did not get any TTSuk2b sequence from the lung sample from Pig 46 from that region. The 12 nt insertion has also been observed previously by other investigators in other TTSuVK2b sequences in NCBI GenBank. One of the closest sequences with the 12 nt insertion was JQ406844-TTSuVk2b/38E05/NZ-2007 from New Zealand, which was 95.32% identical at nt level with our TTSuVk2b sequence. Although the findings of TTSuV1 sequences have been reported in the past from Australia [88], we did not find any previous sequences reported from Australia representing TTSuVk2a and 2b. To the best of our knowledge, our TTSuVk2b sequence is the first report of this genotype from pigs in Australia. Overall, a near identical TTSuVk2b sequence to that found in the Pig 45 lung sample was found in Brazil in 2016 and again in 2018 and was also found in China in 2017.

### 3.9. Analysis of Reads Mapping to Bacteria and to Antimicrobial Resistance (AMR) Genes

Among the detected bacterial species of interest, *Lawsonia intracellularis* was the most abundant and constituted around 2% of all the reads mapped to bacterial nucleic acid in the colon and lung of Pig 45 (2.05–2.08%) and 7.2% of the bacterial reads in the colon of Pig 46. Similarly, *Brachyspira intermedia* constituted around 0.12 to 0.59% of total bacterial reads and *Brachyspira hyodysenteriae* constituted around 0.05 to 0.11% of total bacterial reads with reads matching nearly 100% identical with the chromosome or specific plasmids of these bacteria. Similarly, very few reads mapping to the corresponding bacterial ribosomal RNA were found in these pig samples. Moreover, DNA reads (0.06–0.5% of total bacterial reads) representing *Campylobacter* spp. were also obtained (Appendix A). The high abundance of reads mapped to *Lawsonia intracellularis* was consistent with the histopathology findings reported independently by a commercial diagnostic laboratory with intracellular organisms positive on a Warthin–Starry stain and tissue changes consistent with *Lawsonia intracellularis* enteritis observed by a veterinary pathologist. Bacterial culture isolated a non-haemolytic *Brachyspira* spp. and was unable to isolate any *Salmonella* spp. This was also consistent with what we observed in the bacterial read mapping, identifying reads belonging to *Brachyspira* spp. but not to *Salmonella* spp.

A low number of NGS reads could be assembled into short contigs representing AMR genes, specifically the tetracycline resistance genes *tet* (W) and *tet* (Q) at 97–99% identity over 426–1200 nucleotides and the transposon-mediated lincosamide nucleotidyltransferase, *lun* (C) gene at 99.5% identity over 388 nucleotides from both pigs were detected except Pig 46 lung sample in which AMR genes were not detected. The low abundance of reads and very short contigs assembled of these AMR genes unfortunately did not allow us to assess which bacteria carried these AMR genes, only that these genes were present in one or more species of the microbial community associated with the pigs. Interestingly, the pigs had been treated with drugs within both of these classes of antibiotics in an attempt to control the clinical signs. We found a relatively high abundance of NGS reads for bacterial chromosomal or plasmid DNA for both *Lawsonia intracellularis* and *Brachyspira* spp. compared to their respective ribosomal RNAs, which may indicate that these bacteria may have been dead or inactive possibly as a result of the antibiotic treatments given, and therefore were most likely not the species harbouring the resistance genes.

## 4. Discussion

Porcine enteric diseases like diarrhoea or other intestinal problems and poor growth performance are some of the major problems in pig herds causing huge economic losses around the world [1,2,3,4,5,6,7,8]. In addition, control of enteric disease in pigs is a major reason why antimicrobials are used on farms. Like in our previous study in dogs [44], here NGS was found to be an effective non-targeted technique for the detection and characterisation of a diverse group of viruses and bacteria in pig colon and lung samples. The method was able to identify and quantify a relatively high abundance of the bacterium *Lawsonia intracellularis*, which was initially suspected by the veterinarian and subsequently identified by the histopathologist and could be consistent with the clinical disease, poor growth, and lesions observed [25,26,27]. Furthermore, it could also detect the presence of several species of *Brachyspira* and *Campylobacter*, which may have also been involved in the diarrhoea [28,29,30,31] and also demonstrated that Salmonella was not present in the samples similar to the microbiological testing. Moreover, the NGS method was able to identify several viruses that may have also played a role in the clinical presentation, and enabled the characterisation of these viruses, many for which this study represents the first time they have been genetically characterised in Australian pigs. The method was also able to detect several viruses in very low abundance (e.g., several of the sapelovirus and enterovirus sequences), which suggested that these viruses were not actively replicating at the time of sample collection. These viruses may have infected the pigs at a younger age, and as they were non-enveloped, were likely stable and present in the pigs’ environment.

The pig samples had 0.07–1.37% of the total sequenced reads mapping to different viruses which individually or together (and in combination with the bacteria detected) may have been involved in causing clinical disease in the pigs. Porcine picornaviruses like PSV, PEV-G, and PTV have been isolated from healthy pigs from different countries [50,67,89,90,91,92,93], but were also found to be associated with diseases like diarrhoea, polioencephalomyelitis, respiratory distress, and pneumonia [11,12,15,49,94,95,96,97]. Similarly, PAstVs have been found in both clinically healthy and diarrheic pigs [98,99]. Experiments performed in seven-day-old piglets to study pathogenesis of PAstV have shown clinical manifestations of mild diarrhoea, growth retardation, and damage of the villi of the small intestinal mucosa of piglets after PAstV infection [100]. In our study, the picornaviruses and astroviruses detected showed a high level of genetic diversity when compared with other such viruses identified and sequenced internationally, which may be the result of a high evolution rate together with selection and possibly recombination [101,102,103]. Moreover, multiple genotypes of picornaviruses were detected, particularly for EV-G and PTV, and at least two genetically distinct PoAstV4 in the Pig 46 colon sample were also detected, showing the possibility of future recombination between different EV-Gs genotypes [104], between different PTVs [105], and also between the different PAstV4 genotypes [48], which may give rise to genetically diverse respective virus genotype.

Porcine single stranded DNA viruses like PBoV, PCV2, and TTSuV have also been found in clinically healthy pigs [106,107,108,109]. However, in a study from Germany, PBoV was also found to be associated with coughing, growth retardation, diarrhoea, and encephalomyelitis [110]. Experimental inoculation of six-week-old healthy pigs with PCV2 has been associated with PCV2- systemic disease (PCV2-SD) displaying weight loss [111]. PCV2 in addition to PBoV, PPV, TTSuV, and other viruses have been found to be associated with the development of PMWS, PDNS, and PRRS, causing huge economic losses in the pork industry worldwide [20,40,107,112,113,114]. Interestingly, in this study PCV2 was found, although in low abundance. It is possible that PCV2 infection played some role in the observed diarrhoea and poor growth in conjunction with the other pathogens. It has been shown that pigs affected with PCV2-SD develop severe lymphocyte depletion [115] and hence immunosuppression leading to secondary infection by other pathogens [116]. In this study, lymph nodes were not collected during the initial necropsy to look specifically for lymphoid depletion, however based on our communication with the veterinarian, this farm initiated PCV2 vaccination after this disease event, and subsequent batches of pigs have not suffered from the same performance issues. Similar responses have been observed by other investigators [117,118], and the response perhaps provides some empirical evidence to suggest that PCV2 may have played some role in the clinical disease in this herd before vaccination. Like other studies, parvoviruses in our study showed an intermediate level of genetic diversity [119,120], however the PBoV3 sequences obtained from Pig 46 colon was highly different from those overseas, particularly the VP1 capsid region which may give rise to genetically distinct virus genotypes possibly by recombination events as shown by other studies [77,80].

While most of the virus sequences identified here indicated some divergence from similar viruses overseas, most likely due to geographical isolation of the Australian pig herd and the policy of no imports of live pigs or semen over the last 40 years or so [121,122], surprisingly, we found in the lung sample of Pig 45 a near identical (99.92% and 99.78% identity at nt level) TTSuVk2b sequence to sequences MT671970 and MN272088 reported from Brazil in 2018 and 2016, and 99.86% identical to MK378015 from China in 2017. Similarly, a near identical (99.82% identical at nt level) PCV2b sequence with sequences (DQ220730; Canada-2005, DQ629118; USA-2006, EF565357; Denmark-2007, FJ233907; Canada-2007 and MK504403; USA-2018) reported from Europe and America was obtained in the colon sample from Pig 46, though some studies have suggested a relatively high evolutionary rate among single stranded DNA (ssDNA) viruses like PCV2 [123,124]. Our TTSuVk2b and PCV2b virus sequences were analysed in-depth to check if the obtained virus sequences were a result of laboratory reagent contaminants through, e.g., spin column as shown for another virus [125]. No such TTSuVk2b and PCV2b reads were obtained from human and bird samples processed on the same NGS chip as our pig samples. Moreover, we did not detect any reads of these virus sequences from a nuclease free water negative control that was processed using the same protocol. Interestingly, by retrospectively looking at some NGS results from pig slaughterhouse pooled materials collected by our laboratory from South Eastern Australia in 2017, we were also able to detect the presence of a similar but not identical TTSuVK2b virus, albeit at low levels in the sample. Interestingly, this slaughterhouse sample had a relatively high abundance of PCV2, albeit different to the PCV2 sequenced in this study. The PCV2 (MZ328311-PCV2-ORF1-AUS-2017 and MZ328312-PCV2-ORF2-AUS-2017) obtained from pig samples from the slaughterhouse was 99.8% identical to a PCV2 sequence (EU886638) from Australia in 2007 [126] both in the ORF1 and ORF2 region but 99% similar in the ORF1 and only 91.6% similar in the ORF2 region of the PCV2 sequence obtained from pigs in this study. It is therefore highly likely that these viruses truly came from the samples taken from the Australian pigs, and given their high identity to some viruses overseas, they possibly have a different pattern of introductions into Australia compared to the other viruses, which showed more evidence of independent evolution to related viruses sequenced elsewhere in the world. As these circular ssDNA viruses have been shown to have high levels of environmental stability [127,128,129,130], it is possible that these viruses may have been inadvertently spread internationally, including to Australia, through trade of non-animal products such as imported minerals, vitamins, other feed additives, or via transport containers. Such spread of infectious agents has been observed overseas, i.e., the spread of porcine epidemic diarrhoea virus (PEDV) in North America in 2013 [131,132].

Taken together, the association of these several bacteria and viruses with disease pathogenesis as shown by various studies fits well with the clinical signs of disease observed in the herd in this study and could potentially support the role of these identified pathogens for disease pathogenesis [26,48,94,133,134] in which the viral infection might have followed the bacterial infection or vice versa. Despite being a relatively expensive and time intensive technology, in particular for data analysis of even a single sample, NGS has become a powerful tool and has demonstrated numerous advantages over conventional targeted detection technologies by detecting co-infections, novel or unique and emerging pathogens for which detection methods do not exist. The sensitivity of the technique is very high, and it was able to detect some virus sequences at very low abundance, which indicated that some viruses were likely simply present in the environment of the pigs rather than actively replicating at the time of sampling. Nevertheless, the detection of these low abundance viruses is still useful as it provides information as to potential infections occurring in the pigs earlier in their lives. Overall, we do show here that metagenomic surveillance has the potential to be a very useful monitoring tool to characterize a wide range of pathogens associated with pig production, tease apart pathogen interactions in multifactorial disease, and has the potential to be used as a tool to look for unexpected introductions and transmission of viruses and other micro-organisms between pig farms to assist producers in identifying gaps in biosecurity to prevent the arrival of other pathogens. Further genetic sequencing of more sequences of these viruses from Australian farms will better establish their evolutionary rate and better identify the time frame of their arrival into the Australian pig herd. Moreover, further studies looking at the virus and bacterial pathogens in both healthy pigs and those with poor growth and diarrhoea are needed not only to genetically characterise these diverse potential pathogens but also to clearly understand the detailed pathological and epidemiological roles of the individual pathogens in clinical disease in pigs.

## Figures and Tables

**Figure 1 viruses-13-01608-f001:**
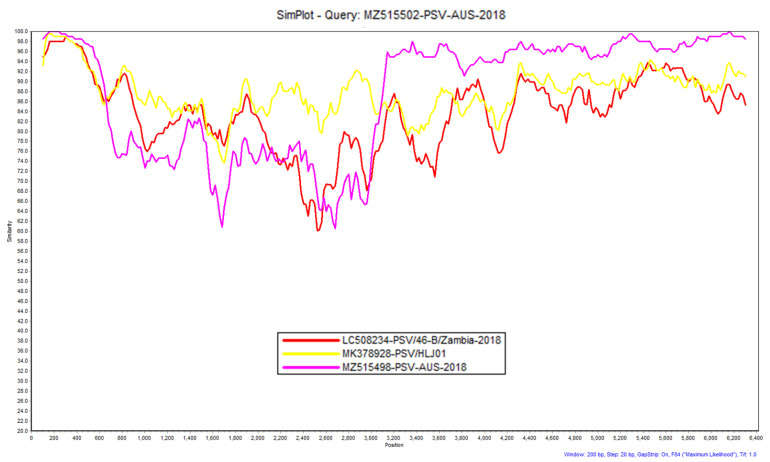
Similarity plot generated in SimPlot using the sequence MZ515502-PSV-AUS-2018 (Pig 46 Colon) as the query sequence against three other sequences using a 200 nucleotide sliding window at 20 nucleotide intervals and the F84 distance [64] model with the maximum likelihood method. Percentage identities at each analysis point were plotted on a line chart. For similarity plot analysis, the *y*-axis shows the percentage similarity between the reference sequences and the query sequence. Different colours are indexed for different reference sequences.

**Figure 2 viruses-13-01608-f002:**
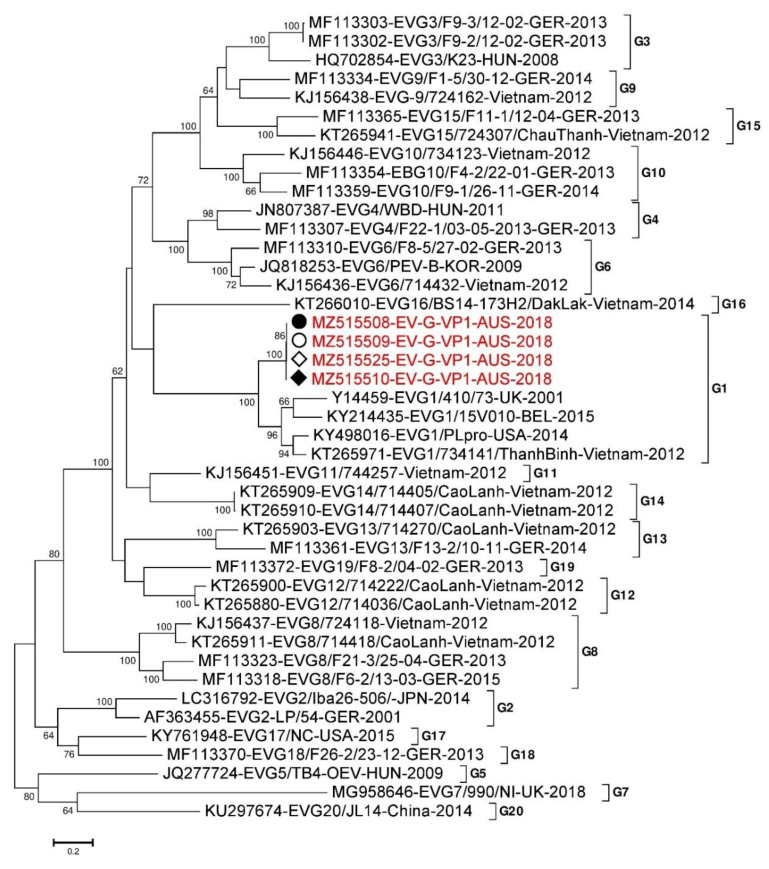
Phylogenetic analysis of complete nucleotide sequence of VP1 capsid region of EV-G. The nucleotide sequences were aligned and analysed using the maximum likelihood method in MEGA 7.0 [63] using the General Time Reversible (GTR + G) [70] model with a bootstrapping of 1000 replicates. The analysis involved 39 reference sequences of the VP1 capsid region of EV-Gs genome and four EV-G sequences from this study. The numbers at nodes represent bootstrap values and values above 60% are shown. Branch lengths are scaled according to the numbers of nucleotide substitutions per site. Sequences name indicated in red colour from Pig 45 colon and lung sample have been labelled with a black (⬤) and white circle (⭘), respectively while Pig 46 colon and lung sample have been labelled with a black (♦) and white rhombus (◊), respectively.

**Figure 3 viruses-13-01608-f003:**
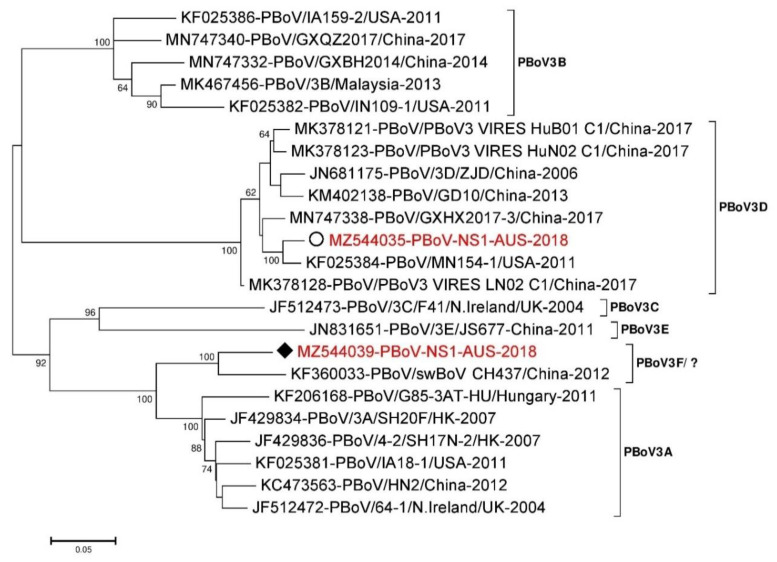
Phylogenetic analysis of the partial nucleotide sequences from the NS1 region of PBoV3. The nucleotide sequences were aligned and analysed using the maximum likelihood method in MEGA 7.0 [63] using the Hasegawa–Kishino–Yano (HKY + G + I) [84] model with a bootstrapping of 1000 replicates. The analysis included 21 reference sequences of the NS1 non-structural region of PBoV3 genome and two PBoV3 sequences from this study. The numbers at nodes represent bootstrap values and values above 60% are shown. Branch lengths are scaled according to the numbers of nucleotide substitutions per site. The sequence name indicated in red colour from Pig 45 lung sample has been labelled with a white circle (⭘) while the Pig 46 colon sample has been labelled with a black rhombus (♦).

**Table 1 viruses-13-01608-t001:** The abundance of virus reads in the pig swab samples. Nucleotide identity percentage range with the closest reference sequence in the structural and non-structural region is shown. The structural and non-structural regions of TTSuV are overlapping and hence sequence identity with the closest reference has been shown in common.

	Abundance of Virus Reads (%)				Identity with Closest Nucleotide Sequence in NCBI at Structural Region	Identity with Closest Nucleotide Sequence in NCBI at Non-Structural Region
	Samples					
	Pig 45, Colon	Pig 45, Lung	Pig 46, Colon	Pig 46, Lung		
	PC45-BC24-AUS-2018	PL45-BC23-AUS-2018	PC46-BC26-AUS-2018	PL46-BC25-AUS-2018		
Virus Name						
Porcine sapelovirus (PSV)	0.3565	0.1805	0.9866	0.0235	~80–88%	~89%
Porcine enterovirus G (PEV-G)	0.2250	0.3537	0.1056	0.0238	~80%	~83–85%
Porcine teschovirus (PTV)	0.0160	-	0.0244	0.0072	~82–86%	~87–89%
Porcine astrovirus (PAstV)	-	-	0.0180	-	~75–81%	~95%
Porcine bocavirus (PBoV)	0.0051	0.0305	0.2246	0.0018	~78–90%	~93–97%
Porcine parvovirus 2 (PPV2)	-	0.0279	-	0.0001	~94%	~95%
Porcine parvovirus 7 (PPV7)	0.0002	-	0.0038	0.0041	~98%	~99%
Porcine bufa virus (PBuV)	0.0068	-	0.0022	0.0007	~94–99%	~99%
Adeno associated virus (AAV)	0.0013	0.0057	0.0023	0.0002	~80–97%	~84–88%
Porcine circovirus 2 (PCV2)	0.0002	0.00007	0.0007	0.00003	99.82%	100%
Torque teno sus virus k2a (TTSuVk2a)	0.0024	0.0119	-	0.00047	~97% (overlapped)	
Torque teno sus virus k2b (TTSuVk2b)	0.0058	0.0249	-	0.013	99.92% (overlapped)	
Total	0.6193	0.6351	1.3682	0.0749		

## Data Availability

With accession numbers (MZ515498–MZ515550, MZ544028–MZ544063). The sequence reads of our porcine circovirus 2 (PCV2) have been deposited in the NCBI Sequence Read Archive (SRA) under SRA accession: PRJNA745489. The fasta file used for PCV2 mapping is available in Appendix A. Additional datasets analysed in the paper can be made available from the authors upon reasonable request.

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
