# Peer review of "Exploring the Cause of Diarrhoea and Poor Growth in 8–11-Week-Old Pigs from an Australian Pig Herd Using Metagenomic Sequencing"

_viruses, 2021, doi:10.3390/v13081608_

Round 1

Reviewer 1 Report

Dear Authors,

the manuscript here presented is well written and the results are clearly reported. The results are very interesting especially when exploring the cause of a symptomatic event as diarrhoea. In addition, the understanding of how pathogens such as viruses and bacteria work synergistically will be useful in the future to prevent economic loss globally.

Material and methods

2.2: The protocol doesn't seem that efficient. Since the protocol used for viruses enrichment produced only the <1.37% of viral reads, the remaining reads were host reads (genomic, ribosomal, etc..)? Is it possible to improve the enrichment?

Reviewer 2 Report

This manuscript attempts to explorer the several causative agents of diarrhea and poor growth by using NGS with lung and colon samples from affected pigs in Australia. The authors found a wide diversity of DNA and RNA viruses as well as bacteria in both lung and colon samples. Based on these observations, the authors may want to propose that these microbes were causative agents for diarrhea and poor growth. However, the information presented in this manuscript does not provide the relation between the microbes and the resulting symptoms. Although the manuscript needs to be modified, it is still informative to understand molecular epidemiology of microbes among pig populations in Australia.
